# Equine Muscle Derived Mesenchymal Stem Cells Loaded with Water-Soluble Curcumin: Modulation of Neutrophil Activation and Enhanced Protection against Intracellular Oxidative Attack

**DOI:** 10.3390/ijms24021030

**Published:** 2023-01-05

**Authors:** Thierry Franck, Justine Ceusters, Hélène Graide, Ariane Niesten, Julien Duysens, Ange Mouithys Mickalad, Didier Serteyn

**Affiliations:** 1Centre of Oxygen Research and Development (CORD), University of Liege, 4000 Liege, Belgium; 2Research Unit FARAH, Department of Clinical Sciences, Faculty of Veterinary Medicine, University of Liege, 4000 Liege, Belgium

**Keywords:** mesenchymal stem cells, curcumin, NDS27, neutrophils, ROS, myeloperoxidase, neutrophil extracellular trap

## Abstract

We investigated the antioxidant potential of equine mesenchymal stem cells derived from muscle microbiopsies (mdMSCs), loaded by a water-soluble curcumin lysinate incorporated into hydroxypropyl-β-cyclodextrin (NDS27). The cell loading was rapid and dependent on NDS27 dosage (14, 7, 3.5 and 1 µM). The immunomodulatory capacity of loaded mdMSCs was evaluated by ROS production, on active and total myeloperoxidase (MPO) degranulation and neutrophil extracellular trap (NET) formation after neutrophil stimulation. The intracellular protection of loaded cells was tested by an oxidative stress induced by cumene hydroperoxide. Results showed that 10 min of mdMSC loading with NDS27 did not affect their viability while reducing their metabolism. NDS27 loaded cells in presence of 14, 7 µM NDS27 inhibited more intensively the ROS production, the activity of the MPO released and bound to the NET after neutrophil stimulation. Furthermore, loaded cells powerfully inhibited intracellular ROS production induced by cumene as compared to control cells or cyclodextrin-loaded cells. Our results showed that the loading of mdMSCs with NDS27 significantly improved their antioxidant potential against the oxidative burst of neutrophil and protected them against intracellular ROS production. The improved antioxidant protective capacity of loaded mdMSCs could be applied to target inflammatory foci involving neutrophils.

## 1. Introduction

In response to inflammation, mesenchymal stem cells (MSCs) are known to migrate to tissue injury sites to participate in immune modulation, tissue remodelling and wound healing [1]. MSCs have been shown to regulate the function of various innate immune cells such as neutrophils [2]. Studies have shown that the modulation of the redox environment and oxidative stress by MSCs can mediate cytoprotective properties and may offer potential antioxidant and anti-inflammatory mechanisms for MSC therapies. In the context of a future therapeutic use, we developed a minimally invasive method to produce equine mesenchymal stem cells from muscle micro-biopsy: muscle derived mesenchymal stem cells (mdMSCs). These cells have similar properties to MSCs from other sources, such as adipose tissue and bone marrow, but these two sources involve a more invasive and difficult sampling [3]. Additionally, we adapted the protocol to minimise interference from the cell culture medium before in vitro and in vivo uses [4]. We showed that equine mdMSCs detached by trypsinization, then washed several times with sterile phosphate buffer saline (PBS), retain an important capacity to inhibit the oxidant response of neutrophils, particularly by modulating the activity of myeloperoxidase [4].

Myeloperoxidase (MPO) is a pro-oxidant enzyme found in the primary granules of neutrophils. When active, MPO can couple H_2_O_2_ with halides/pseudohalides to catalyse the formation of powerful reactive oxygen intermediates as hypochlorous acid, which causes protein, lipid and DNA oxidation and cellular damage [5,6]. MPO also plays a crucial role in the intracellular microbicidal system of phagocytes while associating with chromatin filaments released by neutrophils to form neutrophil extracellular trap (NET) and helps to kill bacteria [7]. During excessive inflammation, the significant release of MPO into the extracellular environment via its degranulation, or NET formation, constitutes a major driver of inflammatory pathologies [5,8]. NET has appeared in numerous reports of diseases and morbidity. Thrombosis mechanisms were reported in COVID-19 patients due to circulating NET formation, including increased MPO-DNA complexes [9].

Priming approaches to empower MSCs have been tried with various cytokines, hypoxia, or pharmacological agents to increase their anti-inflammatory and regenerative properties [10]. Moreover, the homing properties and low immunogenicity of MSCs are two key factors for stem cells used as drug delivery vehicles. The rallying of stem cells to tissue damage and tumor sites is the most important reason for their use in targeted drug delivery [11,12].

More and more studies cover the use of polyphenols together with stem cells in targeting therapeutic applications. Curcumin and curcumin formulations are suitable candidates [13]. Previous studies have shown that curcumin enhanced the regenerative potential of MSCs by: (i) modulating their differentiation; (ii) increasing their viability, cytokine and growth factors secretions; (iii) reducing oxidative stress; and (iv) regulating the neutrophil apoptosis with beneficial effects in several diseases [10,14,15].

Curcumin (diferuloylmethane), a polyphenolic lipophilic molecule from *Curcuma longa* [16,17] exhibits a wide range of biological activities. These include anti-inflammatory, anti-angiogenic, anti-oxidant, wound healing and anti-cancer properties [18]. Of particular interest is the protective activity of curcumin against intracellular ROS production, its free radical scavenging activity and its myeloperoxidase (MPO) inhibitive effect [19,20,21]. However, the major disadvantages of curcumin are its poor water solubility and low systemic bioavailability [22]. Various strategies have been studied to increase its solubility and bioavailability. These include its incorporation in nanoparticles, liposomes, micelles, or phospholipid complexes [17].

We have developed NDS27, a highly water-soluble lysine salt of curcumin (curcumin lysinate), incorporated into hydroxypropyl-β-cyclodextrin (HPβCD). Like curcumin, NDS27 inhibits the activities of both NADPH oxidase [23] and MPO [24] and the oxidative burst of neutrophils. A synergic effect of curcumin and HPβCD was observed on the inhibition of NADPH oxidase assembly compared to curcumin [25]. In vivo, we observed that NDS27 reduced the MPO release by neutrophil degranulation and MPO activity in the broncho-alveolar lavage fluid of horses with LPS-induced lung neutrophilia [26,27]. Loading equine mesenchymal stem cells from microinvasive muscle biopsies (mdMSC) with curcumin or NDS27 has previously been studied. However, the two drugs were added to the culture medium and incubated for at least 2 h before their elimination by washing and cell detachment [28].

In the present study, the objective was to quickly load mdMSCs with curcumin after their detachment by trypsinization and suspension in sterile buffer. Cells were incubated for several minutes with several NDS27 concentrations before their washing and the analysis of the intracellular content of curcumin. Cell viability and metabolism were measured after loading. The anti-inflammatory capacity of loaded cells was evaluated in terms of ROS production, as well as the active and total myeloperoxidase released by neutrophil and attached to the NET. The protective effect of loaded cells was evaluated based on the intracellular oxidative stress induced by cumene hydroperoxide.

## 2. Results

### 2.1. NDS27 Cell Loading, Cell Viability and Metabolism after Loading

The kinetics of NDS27 incorporation was studied with a 14 µM loading concentration, for four incubation times (2, 5, 10, 20 min). The loading was performed with one million cells (in 1 mL HBSS) obtained from the microbiopsies of two horses. Table 1 shows that whatever the incubation time, the percentage of curcumin absorbed by the cells reached nearly 21%, equivalent to ±3 µM.

Based on the kinetic study, a loading time set to 10 min was selected to test several concentrations of NDS27 (14, 7, 3.5, and 1 µM). The concentration of curcumin absorbed by the cells increased according to the dose. Table 2 shows that the percentage of curcumin absorbed by the cells was around 20% for 3.5 and 7 µM of NDS27 and reached 25% with 14 µM NDS27 to obtain an absorbed concentration of ±3.3 µM that confirms the results presented in Table 1. The lowest absorption percentage was observed with 1 µM NDS27 suggesting that the NDS27 concentration was below the saturation condition for optimal cell loading.

The cell viability after a loading time of 10 min was not significantly affected in non-loaded or HPβCD (70 µM) or NDS27 (7 and 14 µM) loaded cells (Table 3). A non-significant slight decrease of viability (±6%) was observed for cells loaded with 14 µM NDS27 in comparison to non-loaded cells. The absolute values for non-loaded cells and cells loaded with NDS27 (14 µM) were 87.4 ± 9.9% and 82.3 ± 11.6%, respectively. By using the MTS assay, we checked the cell metabolism of non-loaded and loaded mdMSCs (Figure 1). Four hours after loading, a decrease in metabolism was observed for loaded cells with NDS27. This decrease was significant with 7 (±27%; *p* < 0.001) and 14 µM NDS27 (±45%; *p* < 0.001).

### 2.2. Effect of NDS27 Loaded mdMSCs on the Stimulated Neutrophils: ROS Production, Total and Active MPO Release and NET Formation

#### 2.2.1. ROS Production

The stimulation of PMNs induced intense ROS production in comparison with non-activated PMNs (Figure 2). The addition of non-loaded and loaded MSCs with HPβCD at a PMN:mdMSCs ratio of 4:1 similarly, and significantly, inhibited ROS production by 29 %, in comparison with the PMNs stimulated alone (*p* < 0.001). When cells loaded with NDS27 were mixed with PMNs, the inhibition of ROS production was amplified in comparison to non-loaded and loaded cells with HPβCD. For cells loaded in the presence of 3.5, 7 and 14 µM NDS27 the inhibitory percentages reached 51%, 67% and 79%, respectively, and was significant (*p* < 0.001) compared to non-loaded cells (HBSS). For 1 µM NDS27, the inhibition reached 28%, but was not significant.

#### 2.2.2. Total and Active MPO Release

Together with an important ROS production, the stimulation of neutrophils was characterized by a release of MPO, which was measured for total and active MPO release by ELISA and SIEFED, respectively (Figure 3). The total MPO release was significantly increased (19.4%; *p* < 0.01) by the stimulated PMNs (Figure 3A, A versus NA). Loaded and non-loaded mdMSCs did not modify release of total MPO by stimulated neutrophils.

For active MPO, the results were quite different (Figure 3B). Stimulation of PMNs induced an important release of the active enzyme. This activity was powerfully inhibited when the PMNs were incubated with mdMSCs (HBSS). This inhibition was more intense and dose-dependent with NDS27-loaded cells (NDS27). Compared to unloaded cells (HBSS), the inhibition was significant for loaded cells with 3.5, 7 and 14 µM NDS27 (*p* < 0.001). HPβCD-loaded cells had a similar effect to non-loaded cells.

#### 2.2.3. NET-MPO Formation

The NET released by stimulated PMNs was measured through its capture by specific antibodies against citrullinated histones followed by the measurement of the bound MPO (total and active) to the NET. The total NET-bound MPO release was significantly increased (about 50%, *p* < 0.001) by stimulated PMNs (Figure 4A, A versus NA). Weak, but significant (*p <* 0.01) inhibition of total NET-bound MPO release was observed for PMNs mixed with non-loaded mdMSCs (HBSS). This inhibition was not modified with HPβCD-or NDS27-loaded mdMSCs.

The active NET-bound MPO release was significantly inhibited when the PMNs were incubated with mdMSCs (HBSS versus A, Figure 4B), and was still more inhibited dose-dependently with the NDS27-loaded cells. Compared with unloaded cells, this inhibition was significant for loaded cells with 3.5, 7 and 14 µM NDS27 (*p* < 0.001).

HPβCD loaded cells had a similar effect to non-loaded cells (HBSS) for total or active NET-bound MPO (Figure 4A,B: HPβCD versus HBSS).

### 2.3. Protective Effect of NDS27 against Intracellular ROS Production

Non-loaded and loaded cells were treated with cumene hydroperoxide for 1 h before the measurement of ROS production by chemiluminescence using L-012 probe.

When cells were treated by cumene, a significant increase of ROS production was observed in comparison with non-treated cells (Figure 5). This ROS production was not affected by cells loaded with 70 µM HPβCD but a strong dose-dependent inhibition of ROS production was observed with NDS27-loaded cells: ±80% inhibition with 3.5, 7 and 14 µM NDS27 (*p* < 0.001), and ±60% with 1 µM NDS27 (*p* < 0.001).

## 3. Material and Methods

### 3.1. Chemicals and Reagents

Analytical-grade phosphate salts, sodium and potassium chloride, sodium hydroxide, sodium acetate, H_2_O_2_ (30%) and Tween 20, Percoll (Cytiva, Boston, MA, USA), T-flasks, conical bottom centrifuge tubes, DMSO, Methanol, Acetonitrile, Formic acid were all purchased from Merck (VWR International, Leuven, Belgium). The 8-amino-5-chloro-7-phenylpyrido [3,4-*d*]pyridazine-1,4(2H,3H)dione (L-012) was purchased from Fujifilm Wako Chemicals Europe GmbH (Neuss, Germany). The bovine serum albumin fraction V (BSA) was obtained from Roche Diagnostics (Mannheim, Germany). Phorbol 12-myristate 13-acetate (PMA), cytochalasin B, N-formyl-methionylleucyl-phenylalanine (fMLP), sodium nitrite, Cumene hydroperoxide (C_6_H_5_-C(CH_3_)_2_-OOH) were purchased from Sigma-Aldrich (Bornem, Belgium). The 96-well microtiter plates (Combiplate 8 EB) and 96-well white plates, the fluorogenic substrate, Amplex red (10-acetyl-3,7-dihydroxyphenoxazine) (Invitrogen), trypsin TrypLE Express (Gibco, Thermo Fisher Scientific, Waltham, MA, USA) and Hank’s balanced salt solution (HBSS) 1 (Gibco, Thermo Fisher Scientific, Waltham, MA, USA) and Fetal Bovine Serum (FBS) were purchased from Fischer Scientific (Merelbeke, Belgium). The Dulbecco’s Modified Eagle Medium Ham’s F12 (DMEM F12) culture medium with Hepes and glutamine, penicillin-streptomycin, amphotericin B and the Dulbecco’s phosphate buffer saline (DPBS) were purchased from Lonza (Verviers, Belgium). The equine MPO ELISA kit was purchased from BiopTis (Vielsalm, Belgium). The purified equine neutrophil MPO was obtained as previously described [29] with the following characteristics: 70.4 U/mg as specific activity and 3.38 mg/mL as protein concentration. The rabbit and guinea pig antibodies against equine MPO were purchased from Bioptis (Vielsalm, Belgium). The rabbit polyclonal antibodies to citrullinated Histone H3 (citrulline R2 + R8 + R17) were purchased from Abcam (Cambridge, UK). CellTiter 96 AQueous One Solution Cell Proliferation Assay was purchased from Promega (Promega Benelux, Leiden, The Netherlands).

NDS27 the lysine salt of curcumin (curcumin lysinate), incorporated into hydroxypropyl-β-cyclodextrin (HPβCD) was provided by BiopTis (Vielsalm, Belgium). The main excipient of NDS27 2-hydroxypropyl-beta-cyclodextrin (HPβCD) was purchased from Roquette SA (Lestrem, France).

### 3.2. mdMSC Culture

The equine skeletal mdMSCs obtained from muscle microinvasive biopsies [3], were provided by Revatis (Aye, Belgium). They were cultured at 37 °C and 5% CO_2_ in Dulbecco’s modified Eagle (DMEM) F-12 culture medium supplemented with 20% heat-inactivated fetal bovine serum (HI-FBS), 100 IU/mL of penicillin-streptomycin and 0.5% of amphotericin B. All experiments have been conducted with mdMSCs between passages five to eight according to the recommendations of Revatis to maintain their capacity for differentiation into adipocytes, osteoblasts, and chondrocytes.

Cells were seeded in 175 cm^2^ culture flasks. For each experiment, cells from 4 T-flasks were generally used. At 75–80% confluency, the medium was removed, and cells were rinsed with 5 mL Dulbecco’s phosphate-buffered saline solution (DPBS). Then, 4 mL of synthetic trypsin (Tryple E Express; Gibco, Grand Island, NY, USA) was added and incubated 5 min at 37 °C. After incubation, 6 mL of DPBS were added and the detached cells were transferred into 50 mL tube. These were centrifuged (330× *g*, 10 min, 37 °C). The cell pellet was resuspended in 5 mL of HBSS and the cell count was determined. The volume of cell suspension (1 million cells/mL HBSS) was adapted to the objective of the experiment for the loading with NDS27 or HPβCD.

### 3.3. Cell Loading with NDS27 and HPβCD

NDS27 is a complex of curcumin lysinate and HPβCD in a 1:5 ratio [30]. NDS27 was dissolved in distilled H_2_O [24]: the starting stock solution was 7.0 × 10^−3^ M in curcumin and serial dilutions were performed in distilled H_2_O to obtain NDS27 solutions containing 1.4, 0.7, 0.35 or 0.1 × 10^−3^ M curcumin.

HPβCD the main excipient of NDS27 being in a 5-fold stoichiometric excess was tested for the highest corresponding concentration of curcumin. HPβCD was dissolved in distilled H_2_O at the starting stock solution of 3.5 × 10^−2^ M. From this solution, one dilution was performed in H_2_O to obtain 7 × 10^−3^ M.

The same volume of each of these dilutions was added to the cell suspension in HBSS to be used at final concentrations of 14, 7, 3.5, 1 µM curcumin and 70 µM HPβCD during the loading. Control samples which did not contain the tested molecules were created by adding distilled H_2_O. Different loading times at 37 °C were also evaluated: 2, 5, 10, 20 min.

After incubation with the drugs, the cell suspensions were centrifuged (330× *g*, 10 min, 37 °C) and the supernatant containing the excess of the drug was removed. The cell pellets were resuspended in the adequate volume of HBSS according to the experimental procedure.

### 3.4. Cell Viability and Metabolism of MSCs after a Loading

Cell viability was evaluated by the Trypan blue exclusion test [31]. Cells were distributed in 5 tubes (1 million cells/990 µL HBSS). In the first tube 10 µL HBSS were added, and the viability was measured directly (T0), by sampling 200 µL of the cell suspension and by adding 50 µL Trypan blue. In the other tubes, the same volume (10 µL) of HBSS, 7 × 10^−3^ M HPβCD or NDS27 (0.7 or 1.4 × 10^−3^ M) was added to the cell suspension and incubated 10 min at 37 °C. After incubation, the cell suspensions were centrifuged (330× *g*, 10 min, 37 °C), the supernatants were removed, and the cells resuspended in 1 mL HBSS. Before the dead cell counting 50 µL of Trypan blue were added to 200 µL cell suspension and 10 µL were placed in a Bürker counting chamber before light microscopy observation.

The cell metabolism was evaluated by the MTS assay by using CellTiter 96 AQueous One Solution Cell Proliferation Assay (Promega). Total absorbance was measured using Microplate Reader at 490 nm. Ten µL of the MTS solution were added to each well containing 100 µL cell suspension. A first absorbance measurement was performed at 490 nm (Microplate Reader, Multiskan, ThermoFischer Scientific) followed by a further measurement 4 h later following plate incubation at 37 °C in darkness. The difference in absorbance was used for cell metabolism expressed in % of control cells without NDS27 loading and without cumene.

### 3.5. HPLC Quantification of Curcumin from NDS27 Incorporated into the Cells

After loading, cell centrifugation, supernatant removal, and pellet suspension in 100 µL HBSS, curcumin was extracted by adding 300 µL methanol. Following incubation for 10 min in ultrasonic bath and centrifugation (10,000× *g*, 10 min, 22 °C), the supernatant was collected and 20 μL injected into the HPLC column (Kinetex 2.6 µm F5 core-shell; 7725i rheodyne injector with a 20 μL loop). Curcumin was eluted at 22 °C with an elution gradient performed with 0.1% formic acid in water (A) and acetonitrile 100% (B) at a flow rate of 1 mL/min (Merck Hitachi L-7100 solvent delivery system). The gradient was set as follows: 0–5 min: 55% B; 6–7 min: 100% B and 8–10 min: 55% B. Eluted compounds were monitored with a Merck Hitachi L-7400 UV detector set at 419 nm, and EZ Chrom ELITE software was used for integration. The data collected by the chromatographic system were analyzed based on the peak area as compared with a standard curve obtained at the concentrations ranging from 0.625 to 10 µM synthetic curcumin.

### 3.6. Effects of Non-Loaded and Loaded mdMSCs on Stimulated Neutrophils: ROS Production, Total or Active MPO Released by Degranulation, and NET Formation

Equine neutrophils used in all experiments were isolated from whole blood (healthy horses, Equine Clinic of Liege University, Sart Tilman Belgium). The neutrophils were isolated according to the method described by Pycock et al. [32] and suspended in DPBS before use. Neutrophils isolated from two horses were spread on microscope slides, stained by Diff-Quik (Medion Diagnostics, Düdingen, Switzerland) and observed by light microscopy (Zeiss, Jena, Germany, Axioskop). Based on five different microscopic fields per horse (100 cells/field), the purity of the neutrophil population was estimated to be 97.1 ± 0.9% neutrophils. The other cells were eosinophils.

#### 3.6.1. Measurement of ROS Production

Dilutions of the cell suspension and reagents were performed to work with a final volume of 200 µL. The superoxide anion production was measured by chemiluminescence (CL) using L-012 probe. Neutrophils (0.5 × 10^6^ neutrophils/well of a white microtiter plate) were incubated for 10 min with 0.125 × 10^6^ mdMSCs non-loaded or loaded with 70 µM HPβCD or 1, 3.5, 7 and 14 µM NDS27. Subsequently, 10 µL of L-012 (1.2 mg/mL in distilled water) and 10 µL of phorbol 12-myristate-13-acetate (PMA) (16 µM in 1% DMSO in ultra-pure H_2_O) were added to obtain with the cells a final concentration of 0.8 µM. After the PMA addition, the CL response was immediately monitored for 30 min and expressed as the integral value of total CL emission. Two controls were performed, one with the PMA activated neutrophils without mdMSCs, and the other with non-stimulated neutrophils and without mdMSCS, where PMA was replaced by its vehicle solution (1% DMSO in H_2_O).

#### 3.6.2. Measurement of Active and Total MPO Release by Neutrophil Degranulation

MdMSCs (0.5 × 10^6^ cells) non-loaded or loaded with HPβCD or NDS27 were mixed in suspension with neutrophils (1 × 10^6^ cells) in a total volume of 1.978 µL of DMEM F-12 medium supplemented with 20% FBS. For control assays, 1 × 10^6^ neutrophils were used without mdMSCs. Cell suspensions were incubated 30 min at 37 °C; 2 µL of a cytochalasin B solution (2 mg/mL in DMSO) were added, except in control assays dedicated to non-activated neutrophils where cytochalasin B was replaced by 2 µL DMSO. Cell suspensions were again incubated 30 min at 37 °C before the addition of 20 µL of a 10^−4^ M fMLP solution (in 10% DMSO ultra-pure water) except in the controls with non-activated neutrophils where fMLP was replaced by 10% DMSO in ultra-pure water. The samples were let incubated for 24 h at 37 °C, then centrifuged (350× *g*, 10 min, 22 °C) and the supernatants collected and frozen at −20 °C until MPO assay.

MPO activity present in a sample was measured by SIEFED technique that we have developed for the specific detection of active neutrophil MPO in complex biological samples. This reveals specifically MPO activity after the enzyme capture by rabbit anti-equine MPO antibodies (3 µg/mL) precoated on a microplate [29].

Hundred µL of the MPO containing the sample were loaded on the SIEFED microplate. Sampling was performed in duplicate. After an incubation for 2 h at 37 °C, the samples were carefully removed, and the plate was drained on a paper towel to remove any remaining liquid. The wells were washed four times with a PBS solution containing 0.1% Tween 20. Finally, the peroxidase activity of the MPO captured by the antibodies was monitored by adding 100 μL of a 40 μM Amplex red solution. This was freshly prepared in a 50 mM phosphate buffer, pH 7.4, supplemented with 10 μM H_2_O_2_ and 10 mM sodium nitrite. The reaction was followed by fluorimetry. Total fluorescence developed during 30 min (37 °C) was monitored using a Fluoroskan Ascent (Thermo Scientific) set at 544 nm and 590 nm for excitation and emission wavelengths, respectively. Total fluorescence was directly proportional to the amount of active MPO present in the sample.

An ELISA assay kit (Equine MPO ELISA, BiopTis, Vielsalm, Belgium) was used to measure the total equine MPO concentration in the collected supernatants, which were diluted 200× with 20 mM PBS containing 0.5% BSA and 0.1% Tween 20.

#### 3.6.3. Measurement of the Active and Total MPO Bound to the NET (NET-MPO)

HPβCD or NDS27 loaded and non-loaded mdMSCs (0.5 × 10^6^ mdMSCs) were mixed with neutrophils (1 × 10^6^ cells) in 1.978 µL of DMEM F-12 medium supplemented with 20% FBS. In control assays, only 1 × 10^6^ neutrophils were added. Cell suspensions were incubated 30 min at 37 °C, then 2 µL of a cytochalasin B solution (2 mg/mL in DMSO) were added. A control assay was dedicated for non-activated neutrophils, where cytochalasin B was replaced by its vehicle solution. Cell suspensions were again incubated 30 min at 37 °C before an addition of 20 µL of a 10^−4^ M fMLP solution (in 10% DMSO ultra-pure water) in all assays. In the control with non-activated neutrophils, fMLP was replaced by 10% DMSO in ultra-pure water. The samples were incubated 24 h at 37 °C. Subsequently, the tubes were centrifuged (350× *g*, 10 min, 22 °C) and the supernatants were collected and frozen at −20 °C before measurement.

The NET released by stimulated neutrophils was captured by anti-histone H3 (citrulline R2 + R8 + R17; anti-H3Cit) antibodies based on the method described by Thälin et al. [33] Transparent 96-wells microplates were coated overnight, at 4 °C, with 100 μL/well rabbit anti-H3Cit antibodies (0.5 μg/mL) diluted with 20 mM PBS buffer. After removal of the coating solution, the plates were incubated (150 min, 22 °C) with 200 µL blocking buffer (PBS buffer with 5 g/L of BSA). These were washed four times with the washing solution (PBS buffer with 0.1% Tween 20). The plates were dried for 3 h at 22 °C, then conserved in a dry atmosphere in a hermetic bag at 4 °C until use. The supernatants dedicated to the NET assay were loaded (100 µL) in duplicate into the wells of the anti-H3Cit antibodies-coated microplate and incubated 2 h at 37 °C. After incubation, the supernatants were removed, and the wells were washed four times with the washing solution. After the NET capture, the presence of active MPO bond to the NET was detected by the SIEFED technique as described above (see Section 3.6.2).

The total MPO bond to the NET was detected by using a second anti-MPO antibody (guinea pig antibody coupled to alkaline phosphatase) and the revelation technique from the ELISA assay kit (see Section 3.6.2).

### 3.7. Intracellular mdMSCs ROS Production under Cumene Attack

Cumene hydroperoxide is an intracellular ROS generator [34]: it was used to submit the mdMSCs to oxidant stress. It is able to induce ROS production by the cells. The lack of direct cytotoxicity by cumene was controlled by the measurement of their metabolism and viability after the cumene treatment (see Section 3.4).

Unloaded mdMSCs and loaded mdMSCS with NDS27 (14, 7, 3.5, 1 µM) or HPβCD (70 µM) were resuspended in 990 µL HBSS and treated with cumene through the addition of 10 µL of a 10^−2^ M cumene hydroperoxide solution in ethanol. A control assay (Ctrl-) was performed with unloaded mdMSCs where cumene was replaced by ethanol. A further incubation (30 min at 37 °C) was performed, then 100 µL of the cell suspension were distributed in triplicate in the wells of white 96 well-microplates for CL measurement and in 96-well UV-transparent microplates for the cell metabolism assay using MTS.

Before measuring ROS production, 100 µL of equine MPO dissolved in DPBS (20 mU/mL) were added to each well. The ROS production was measured by chemiluminescence (CL) with an L-012 probe. Ten µL of a L-012 solution in water (1.2 mg/mL) were added to each well. After 30 min incubation at 37 °C, the CL response was monitored for 30 min at 37 °C (Fluoroskan, ThermoFischer Scientific) and the data were expressed as the integral value of the total CL emission.

### 3.8. Statistical Analysis

Experiments were performed with cell batches from at least three horses for mdMSCs isolation and culture, and from five horses for blood sampling and neutrophil isolation. In some experiments, mdMSCs from the same horse were tested twice with neutrophil batches from two different horses. Each point in the experiment was performed at least in duplicate. Five independent experiments in duplicate were taken into consideration for the figures and statistical analysis. For the figures, data were expressed as mean standard deviation (SD) and given in relative values (%) by reference to control groups taken as 100%. As some populations or data series did not follow a Gaussian distribution, a nonparametric test was used: a two tail Mann–Whitney test was performed (Graph Pad, InStat, San Diego, CA, USA). A *p*-value < 0.05 was considered significant.

## 4. Discussion

We succeeded in the obtention and culture of significant quantities of equine MSCs originating from skeletal muscle micro-biopsies [3] and developed a protocol to minimise the interferences of the cell culture medium in the mdMSCs batches prepared for in vitro experiments [3,4]. We showed that equine mdMSCs submitted to a process involving repeated washings with sterile phosphate buffer saline solution (PBS) maintain significant capacity for inhibition of the oxidant response of neutrophils [4].

In recent years, several priming approaches have been proposed to empower MSC properties. This has given variable results [10,35]. Many studies concern the use of polyphenol in copresence with stem cells for targeting therapeutic application [13]. Curcumin formulations constitute suitable candidates [36,37,38,39,40,41,42]. Indeed, curcumin therapy increasingly seems an attractive alternative in patients with contraindication to non-steroidal anti-inflammatory drugs (NSAIDs) [43]. However, low water solubility, poor absorption, instability in alkaline environment, elimination and rapid metabolism limit the utilization of curcumin [44]. Thus, many derivatives, prodrug and combinations of drug therapy have been studied [44]. Among them, the loading of MSCs with curcumin combined with their tropism towards tissue damage constitutes a major asset for targeting the delivery of curcumin in a restricted inflammatory environment [11,12]. Another well-studied aspect is the improvement of the water solubility of curcumin [45].

Cultured MSCs were generally incubated several hours with native curcumin solubilized in DMSO or encapsulated in nanoparticles or micelles [36,42]. In our study, the cells were loaded for a short time after detachment from the culture flask with a water-soluble form of curcumin NDS27.

### 4.1. mdMSCs Loading with NDS27

We loaded mdMSCs with NDS27, a water-soluble form of curcumin consisting of curcumin lysinate encapsulated in hydroxypropyl-beta-cyclodextrin (HPβCD) [30]. NDS27 is easily solubilized in ultra-pure water and further diluted in an HBSS buffer. The loading of mdMSCs appeared NDS27-dose dependent, reaching around 20% for the two highest NDS27 concentrations to obtain intracellular concentrations of 1.53 µM and 3.38 µM curcumin, respectively (Table 2). Interestingly, the loading was rapid, and 20% loading capacity was reached after 2 min incubation at 37 °C (Table 1). In a previous study, we loaded adherent mdMSCs with NDS27 or native curcumin for 2 h with 42 µM curcumin or NDS27, and the loading yield was only 2%, which corresponded to about 0.8 µM internalized curcumin [28]. Another study with hepatic HuH-7 cells showed that curcumin entered within seconds into the cells and was already maximal after 5 min but only one-twentieth of the curcumin had entered the cells [46,47]. The good performance of our cell loading can be explained by a different cell preparation, that in our case were suspended cells rather than adherent cells used by Colin et al. [28]. Moreover, we performed the loading in an aqueous buffer (HBSS buffer) rather than in the culture medium. It has been shown that proteins from the culture medium such as albumin can interfere with the uptake of curcumin by cells [48]. The high solubility of NDS27 in the aqueous buffer, thanks to the presence of HPβCD, facilitates the exchange of curcumin with cell membrane [23].

For our further experiments, we adopted a 10 min incubation at 37 °C for mdMSCs loading.

### 4.2. NDS27 Effects on mdMSCs Metabolism and Viability

Only a slight decrease of cell viability (±6%) was observed after the cell loading for 10 min. (Table 3). In parallel, we checked the metabolic status of the cells. In viable cells, NAD(P)H-dependent oxidoreductase enzymes of the mitochondria reduce tetrazolium into coloured formazan, proportionally to the metabolic activity of the mitochondria [49]. Results showed a significant decrease for the two highest NDS27 concentrations (Figure 1).

Several studies demonstrated an activity of curcumin on mitochondria. In vitro curcumin mitigated the effects of cellular stress by decreasing the mitochondrial functions, including oxygen consumption, ATP production, calcium retention, and membrane potential [50]. Curcumin protected mitochondria from dysfunction by retaining the mitochondrial membrane potential (ΔΨm) and the activities of all four mitochondrial complexes (complex I, II, III, and IV) [51]. In the study performed by Colin et al. [28], curcumin was localized in mitochondria of adherent mdMSCs loaded with NDS27, and they observed a transient modification of the mitochondrial metabolism which did not persist behind 24 h after cell loading. Our results confirm a potential beneficial role of curcumin on the mitochondrial function of NDS27-loaded mdMSCs, but further studies are needed to enlarge on this. The preconditioning of mdMSCs with curcumin could thus protect the cell against undesirable oxidative stress and assist their regenerative function in inflammatory conditions. Several studies showed that the preconditioning of mesenchymal stem cells with curcumin accelerated wound healing [15,39,52] by promoting cell survival and mitochondrial quality [15].

### 4.3. Protective Effects of NDS27 Loaded mdMSCs against the Oxidant Activity of Neutrophils

Previous studies demonstrated cytoprotective effects of curcumin pre-treatment (10 μmol/L for 24 h) on rat bone-marrow and adipose tissue-MSCs against an oxidative stress induced by an exposure to H_2_O_2_ [14,21]. Xu et al. [53] found that small extracellular vesicles derived from curcumin primed adipose tissue derived MSCs exerted enhanced protective effects against osteoarthritis by inhibiting oxidative stress and chondrocyte apoptosis.

To our knowledge, the antioxidant capacity of curcumin-loaded cells has never been studied on the oxidant response of neutrophils. We previously observed that mdMSCs inhibited the ROS production by PMA-stimulated neutrophils [4]. Here we demonstrated that the loading with NDS27, almost with 3.5, 7 and 14 µM, significantly improved this inhibition compared to non-loaded and HPβCD-loaded cells (Figure 2). In the same context, by using our previously published technique [4], we observed that loaded mdMSCs did not interfere with the release of total MPO (Figure 3A) and NET formation (Figure 4A) by activated neutrophils. However, NDS27-loaded cells inhibited the activity of MPO, whether the enzyme was free or bound to the NET, in comparison with non-loaded or HPβCD-loaded cells (Figure 3B and Figure 4B). The antioxidant and anti-inflammatory capacity of NDS27 and mdMSCs on neutrophil oxidative response, NADPH oxidase activity, and MPO activity has already been separately demonstrated [4,24,25], but we clearly highlighted now that the combination of NDS27 and mdMSCs makes it possible to further improve these protective effects. However, the mechanisms of these anti-oxidative effects need further studies: are they partially related to an increased cell to cell contact, to the release of cell debris or vesicles enriched with curcumin, to an increase of the paracrine activity of the mdMSCs, or even to an in situ synergic antioxidant effect? Previous studies showed that curcumin included into liposomes keeps its antioxidant potential and can limit oxidative and nitrosative stress following ischemia reperfusion injury [54], and that a preconditioning of adipose-derived MSCs with curcumin enhances their paracrine activity and helps to accelerate healing of burn wounds [39]. MSCs loaded with inulin-D-alfa-tocopherol succinate micelles can be also used as a curcumin delivery system for the treatment of neurodegenerative diseases [42].

### 4.4. Protective Effect of ND27 against Intracellular Oxidant Stress on mdMSCs

In addition to its protective effects against the oxidative burst of neutrophils, the loading of mdMSCs with NSD27 affords them the capacity to protect themselves against an intracellular oxidant stress (Figure 5) induced by cumene hydroperoxide, a stable organic oxidizing agent. Its peroxy group -O-O- generates lipophilic radicals by the subtraction of a hydrogen atom from lipid molecules. This initiates a lipid peroxidation cascade [55]. Cumene hydroperoxide can react with amino acids and proteins, producing multiple effects, such as oxidation side-chain unfolding or conformational changes, enzymatic inactivation, and alterations in cellular handling and turnover of proteins [56]. Additionally, it can generate mitochondrial ROS production by acting on complex I and complex III or on complex III alone with or without O_2_ consumption by the mitochondria [57]. Our results obtained with NDS27 are in agreement with Barzegar and Moosavi-Movahedi [19], who showed that curcumin can diffuse through the membrane into the cells, where it prevented the production of ROS required to oxidize 2′,7′-dichlorofluorescin (DCFH2), a probe widely used to detect intracellular ROS.

In the present study, we detected ROS production following cumene hydroperoxide treatment, with the chemiluminescent probe (L012), a stable derivative of luminol that allows a sensitive measurement of the overall ROS production from both intra- and extracellular origins [58]. Overall, our results suggest that loaded cells, through their curcumin content, efficiently reduced ROS production by stabilizing oxidized or radical species produced by stressed cells. Ghufran et al. [59] considered that curcumin priming preserves the survival and growth potential of adipose-derived stem cells under stress conditions.

### 4.5. Conclusions

From our results, the learn lessons are the followings:We have developed a quick and efficient method to load, with a good yield (20%), equine mdMSCs suspensions with the water-soluble curcumin derivative NDS27. The loading decreases the cell metabolism but does not induce cytotoxicity;The loading with NDS27 increases the anti-inflammatory activity of mdMSCs against stimulated neutrophils, as measured by the lowering of ROS production and the significant inhibition of the catalytic activity of degranulated or NET- bound MPO;The loaded mdMSCs acquire a major capacity to decrease the intracellular ROS production induced by an oxidant stress while decreasing cell metabolism.

In conclusion, loading equine muscle mesenchymal stem cells with water-soluble curcumin NDS27 enhances their inhibitory properties against neutrophil oxidative burst and provides them with protection against oxidative attacks.

An outcome of this study should be the potential clinical use of curcumin-loaded MSCs in inflammatory pathologies involving excessive neutrophil activity.

## Figures and Tables

**Figure 1 ijms-24-01030-f001:**
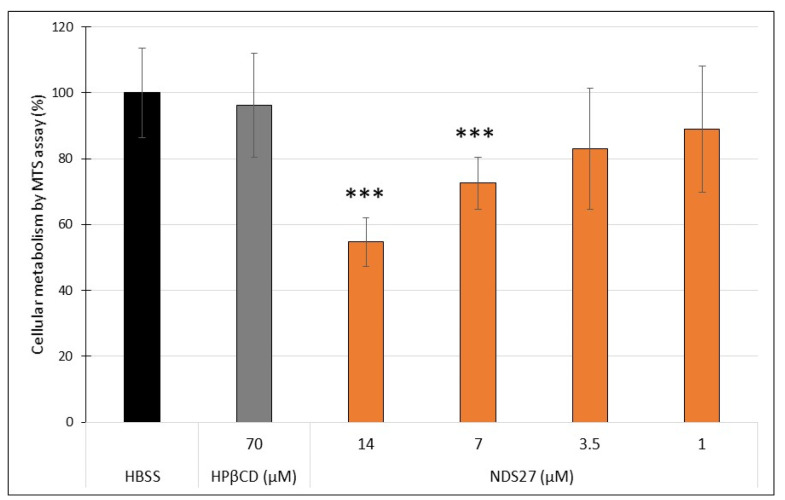
Assessment of cell metabolism of non-loaded (HBSS) and loaded mdMSCs with HPβCD (70 µM) or NDS27 (1, 3.5, 7, 14 µM). The absorbance was read 4 h (T4) after MTS addition (T0) (incubation at 37 °C). Results correspond to the difference of absorbance between T4 and T0. The response of non-loaded cells (HBSS) was set as 100%. Mean ± SD, *n* = 5 horses. *** *p* < 0.001 vs. HBSS.

**Figure 2 ijms-24-01030-f002:**
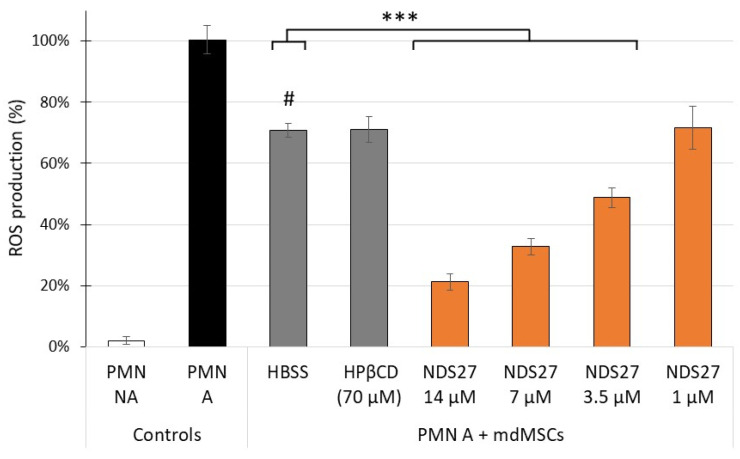
Effect of non-activated (PMN NA) and PMA activated neutrophils (PMN A) alone or mixed with non-loaded (HBSS) and loaded mdMSCs with HPβCD (70 µM) or NDS27 (1, 3.5, 7, 14 µM) in PBS on ROS production. The numbers of neutrophils and mdMSCs per well were 0.5 × 10^6^ and 0.125 × 10^6^ respectively. Means ± SD are shown in relative percentages versus stimulated neutrophils without mdMSCs (PMN A) defined as 100% response. Statistical analysis was considered for 5 independent experiments (2 technical replicates) with MSCs from 3 horses. *** *p* < 0.001 vs. HBSS, # *p* < 0.001 vs. PMN A.

**Figure 3 ijms-24-01030-f003:**
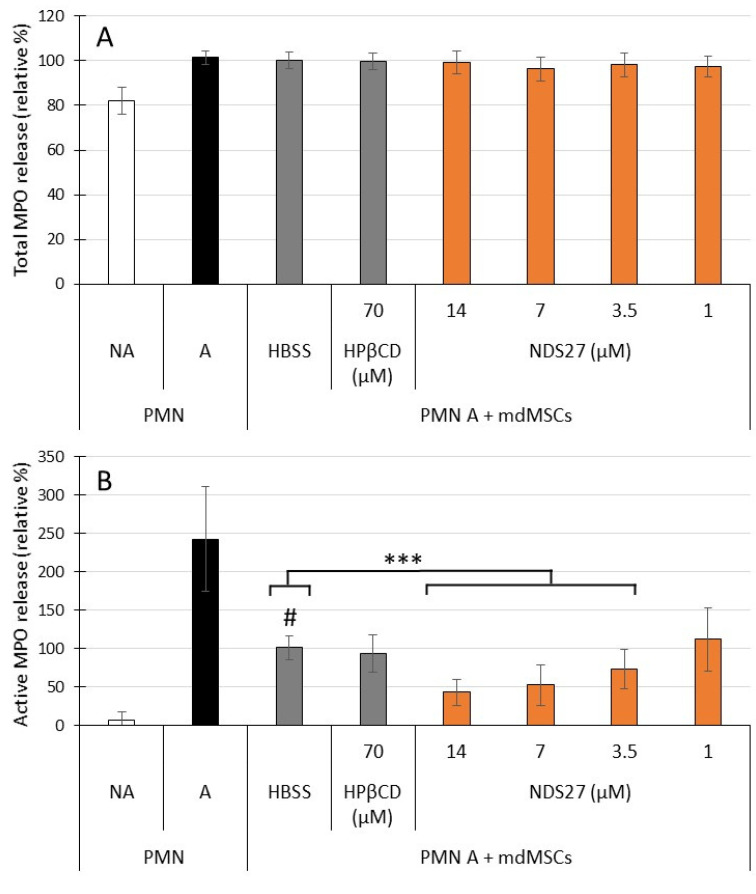
Effect of non-activated (PMN NA) and CB + fMLP activated neutrophils (PMN A) alone or mixed with non-loaded (HBSS) and mdMSCs loaded with HPβCD (70 µM) and NDS27 (1, 3.5, 7, 14 µM) on total (**A**); and active (**B**) MPO released by PMNs. The numbers of neutrophil and mdMSCs per 2 mL medium were 1 × 10^6^ and 0.5 × 10^6^ respectively. Means ± SD are shown in relative percentages versus non-loaded cells (HBSS) in the presence of stimulated neutrophils defined as 100% response. Statistical analysis was considered for 5 independent experiments (2 technical replicates) with mdMSCs from 3 horses. *** *p* < 0.001 vs. HBSS, # *p* < 0.001 vs. PMN A.

**Figure 4 ijms-24-01030-f004:**
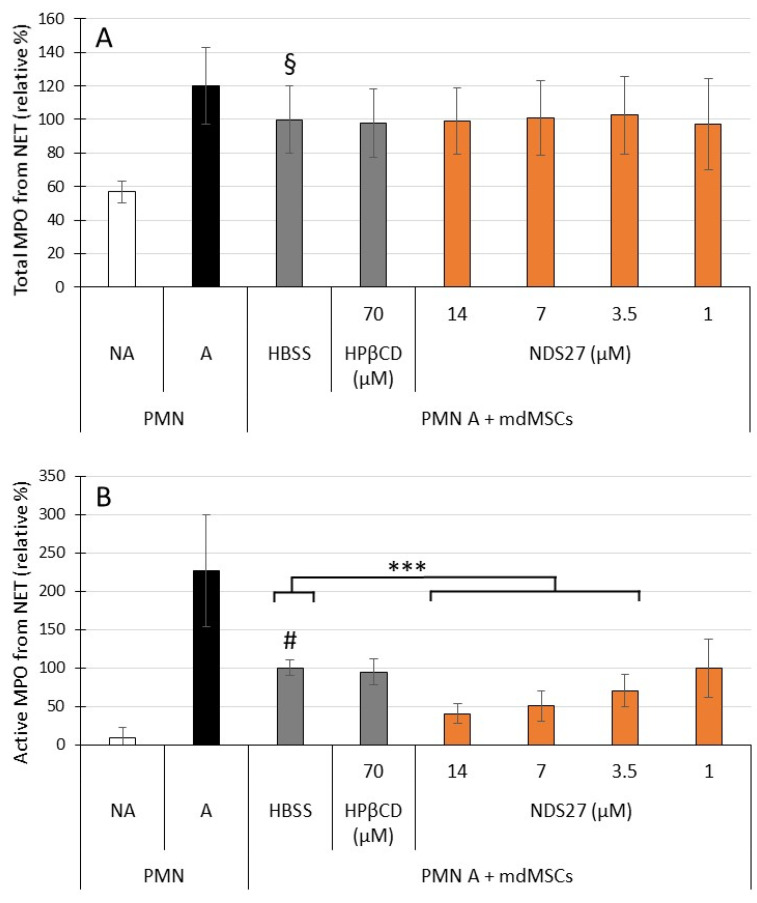
Effect of non-activated (PMN NA) and CB + fMLP activated neutrophils (PMN A) alone or mixed with non-loaded (HBSS) and loaded mdMSCs with HPβCD (70 µM) or NDS27 (1, 3.5, 7, 14 µM) on total (**A**); and active (**B**) MPO bound to the NET. The numbers of neutrophil and MSCs per 2 mL medium were 1 × 10^6^ and 0.5 × 10^6^ respectively. Means ± SD are shown in relative percentages versus non-loaded cells (HBSS) in the presence of stimulated neutrophils defined as 100% response. Statistical analysis was considered for 5 independent experiments (2 technical replicates) with MSCs from 3 horses. *** *p* < 0.001 vs. HBSS, # *p* < 0.001 vs. PMN A, **§**
*p* < 0.01 vs. PMN A.

**Figure 5 ijms-24-01030-f005:**
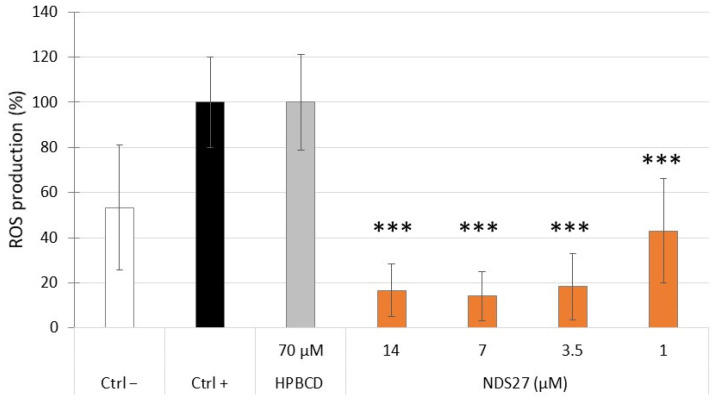
Effect of HPβCD (70 µM) or NDS27 (1, 3.5, 7, 14 µM) loading of mdMSCs on the intracellular ROS production (CL measurement with L-012) after treatment with cumene hydroperoxide. Means ± SD (*n* = 5 horses, 2 technical replicates) are shown in relative percentages versus non-loaded cells treated with cumene (Ctrl+) defined as 100%. Negative control (Ctrl−): non-loaded cells where cumene hydroperoxide was replaced by its vehicle solvent (100% Ethanol). *** *p* < 0.001 vs. Ctrl+.

**Table 1 ijms-24-01030-t001:** Concentration of curcumin absorbed by one million mdMSCs after 2, 5, 10 and 20 min incubation at 37 °C in 1 mL of a 14 µM NDS27 solution (in HBSS). *n* = 3 horses. f: final.

Incubation Time (min)	[Mean Curcumin] f (µM)	SD	CV (%)	Absorbed Curcumin (%)
2	3.12	0.20	6.29	22.3 ± 1.4
5	2.84	0.13	4.62	20.3 ± 0.9
10	3.03	0.23	7.72	21.6 ± 1.7
20	3.03	0.49	16.09	21.7 ± 3.5

**Table 2 ijms-24-01030-t002:** Concentration of curcumin absorbed by one million mdMSCs after 10 min incubation at 37 °C in 4 different NDS27 concentrations (in 1 mL HBSS). *n* = 5 horses. i and f: initial and final.

[Curcumin] i (µM)	[Mean Curcumin] f (µM)	SD	CV (%)	Absorbed Curcumin (%)
14	3.38	0.20	5.81	25.2 ± 1.4
7	1.53	0.16	10.53	22.3 ± 2.3
3.5	0.69	0.11	16.43	18.2 ± 3.2
1	0.13	0.05	40.01	11.5 ± 5.4

**Table 3 ijms-24-01030-t003:** Viability of mdMSCs after their loading with 70 µM HPβCD, or with NDS27 at 7 µM or 14 µM. Non-loaded cells were in the presence of HBSS. Mean ± SD, *n* = 5 horses.

	mdMSCs with	Viability (%)
Before loading	HBSS	87.46 ± 7.86
After loading (10 min)	HBSS	87.39 ± 9.91
	HPβCD (70 µM)	85.82 ± 11.76
	NDS27 (7 µM)	86.87 ± 7.07
	NDS27 (14 µM)	82.26 ± 11.58

## Data Availability

Not applicable.

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
