# Peer review of "Equine Muscle Derived Mesenchymal Stem Cells Loaded with Water-Soluble Curcumin: Modulation of Neutrophil Activation and Enhanced Protection against Intracellular Oxidative Attack"

_ijms, 2023, doi:10.3390/ijms24021030_

Round 1

Reviewer 1 Report

Authors claimed that the loading of mdMSCs with NDS27 significantly improved their antioxidant potential against the oxidative burst of neutrophil and protected them against intracellular ROS production. This antioxidant capacity of loaded mdMSCs could be applied to target inflammatory foci involving neutrophils.

Authors developed NDS27 complex of curcumin lysinate incorporated into hydroxypropyl-β-cyclodextrin which inhibiting both NADPH oxidase (Derochette, FEBS Open Bio 2014) and MPO.

The purpose and method are clear and results are reasonable.

The weakness is the novelty is relatively low. The authors is requested to define what the methods are. Furthermore, what and why the quick and efficiency are. Please address them both in the introduction and discussion.

Is the NDS27 better than other selective anti-inflammatory drugs? Please illustrate.

Please explain why select the mdMSCs instead of other cell candidate.

Author Response

Dear Reviewer,

I would like to thank for your comments and remarks which help to improve the paper.

We hope we have answered your questions sufficiently.

Thank you for your assistance

Thierry Franck 

Reviewer 1

  • The weakness is the novelty is relatively low. The authors is requested to define what the methods are.

Thank you for this suggestion which helps us to improve our paper.

We added several sentences in the introduction to evidence our specificity compared to other studies. We completed the objective in order to introduce the methods used in our study. 

From line 35:  In the context of a future therapeutic use, we developed a minimally invasive method to produce equine mesenchymal stem cells from muscle micro-biopsy: muscle derived mesenchymal stem cells (mdMSCs). These cells have similar properties than MSCs from other sources such as adipose tissue and bone marrow but that involve a more invasive and difficult sampling [3]. Additionally, we adapted the protocol to minimise interference from the cell culture medium before in vitro and in vivo uses [4] . We showed that equine mdMSCs detached by trypsinization then washed several times with sterile phosphate buffer saline (PBS) keep an important capacity to inhibit the oxidant response of neutrophils particularly by modulating the activity of myeloperoxidase [4].

From line 83:  Loading equine mesenchymal stem cells from microinvasive muscle biopsies (mdMSC) with curcumin or NDS27 has previously been studied. However, the two molecules were added to the culture medium, let incubate for at least 2 hours before their elimination by washing and detachment [28].  In the present study, the objective was to load quickly mdMSCs with curcumin after their detachment by trypsinisation and suspension in sterile buffer. Cells were incubated for several minutes with several NDS27 concentrations before their washing and analysis of the intracellular content of curcumin. Cell viability and metabolism were measured after loading. The anti-inflammatory capacity of loaded cells was evaluated in terms of ROS production, as well as active and total myeloperoxidase released by neutrophil and attached to the NET. The protective effect of loaded cells was evaluated based on the intracellular oxidative stress induced by cumene hydroperoxide.

  • Furthermore, what and why the quick and efficiency are. Please address them both in the introduction and discussion.

The better loading efficiency can be explained by the difference between the loading conditions performed with adherent cells or cells in suspension. This was introduced from line 83 (see above). This also allows a better connexion with the beginning of the discussion (line 440 and line 458).

From line 458: The good performance of our cell loading can be explained by a different cell preparation that in our case were suspended cells rather than adherent cells as used by Colin et al. [28]. Moreover, we performed the loading in aqueous buffer (HBSS buffer) rather than in the culture medium. It has been shown that proteins in the culture medium such as albumin can interfere with the uptake of curcumin by cells [48]. The high solubility of NDS27 in aqueous medium, thanks to the presence of HPbCD allows to facilitate the exchange of curcumin with cell membrane [23]. 

We also add an additional reference to explain that NDS27 is more efficient than curcumin for NADPH oxidase inhibition due to a synergic effect of curcumin and HPBCD on the NADPH oxidase assembly. From line 77: Like curcumin, NDS27 inhibits the activities of both NADPH oxidase  [23] and MPO [24] and  the oxidative burst of neutrophils. A synergic effect of curcumin and HPβCD was observed on the inhibition of NADPH oxidase assembly compared to curcumin [25].

  • Is the NDS27 better than other selective anti-inflammatory drugs? Please illustrate.

Additional references were added in the discussion to introduce curcumin as alternative to other anti-inflammatory drugs (NSAIDS). 

From line 432: Indeed, curcumin therapy increasingly seems an attractive alternative in patients with contraindication to non-steroidal anti-inflammatory drugs (NSAIDs) [43]. However, low water solubility, poor absorption, instability in alkaline environment, elimination and rapid metabolism limit the utilization of curcumin [44]. Thus, many derivatives, prodrug and combination of drug therapy are studied [44]. Among them, the loading of MSCs with curcumin combined with their tropism towards tissue damage constitutes a major asset for targeting the delivery of curcumin in a restricted inflammatory environment [11,12]. Another well studied aspect is the improvement of the water solubility of curcumin [45].

  • Please explain why select the mdMSCs instead of other cell candidate.

Bone marrow and adipose tissue represent the two most exploited sources of adult mesenchymal stem cells for musculoskeletal applications. Unfortunately, the sampling of bone marrow and fat tissue is invasive and does not always lead to a sufficient number of cells. In this paper we used cells from non-invasive microbiopsy of skeletal muscle. This was stated in the introduction part from line 35.

Another particularity of mdMSCs and MSCs in general is also their tropism towards damages tissues that make them good candidate for drug delivery. This particularity was also explained in the paper.

From line 55: Priming approaches to empower MSCs have been tried with various cytokines, hypoxia, or pharmacological agents to increase their anti-inflammatory and regenerative properties [10]. Moreover, the homing properties and low immunogenicity of MSCs are two key factors for stem cells to be used as drug delivery vehicles. The rallying of stem cells to tissue damage and tumor sites is the most important reason for their targeted drug delivery [11,12].

From line 434: However, low water solubility, poor absorption, instability in alkaline environment, elimination and rapid metabolism limit the utilization of curcumin [44]. Thus, many derivatives, prodrug and combination of drug therapy are studied [44]. Among them, the loading of MSCs with curcumin combined with their tropism towards tissue damage constitutes a major asset for targeting the delivery of curcumin in a restricted inflammatory environment [11,12].

Reviewer 2 Report

In this manuscript the authors propose identify the possible anti-inflammatory effect of
the Curcumin solution on the neutrophil activation. The activation of neutrophils was
measure by the ROS production as well as MPO release.
 In general the study was good designed and the conclusions are supported by the
results. The methods used are adequate. I think the manuscript is promising and with
the below recommendations can be suitable for publication:

1.    The neutrophil isolated were characterized with any cellular marker?, What is the purity of neutrophils obtained?

2.    The concentration of PMA frequently used are to 10 at 100nM, why the authors used 16uM? This concentration are elevated and could by cytotoxic.

T. Franck, et. at..  doi.org/10.1016/j.vetimm.2009.02.015 report 0.8uM for neutrophils activation

3.    The authors deduce the NET’s formation by elevation of ROS production and the release of MPO but not include any image for make sure of these.

Author Response

Dear Reviewer,

I would like to thank for your comments and remarks which help to improve the paper.

We hope we have answered your questions sufficiently.

Thank you for your assistance

Thierry Franck 

Reviewer 2

  1. The neutrophil isolated were characterized with any cellular marker?, What is the purity of neutrophils obtained?

We isolated neutrophils by the Percoll gradient described by Pycock et al. 1987. This technique allows a very good separation between the different leucocyte layers. We generally made smears after neutrophil isolation to estimate the percentage of the neutrophil in the preparation.  

Please find attached in the figure below a picture summarising the preparation of neutrophils isolation and photos made after a diff-quick staining of the neutrophils smear. The picture A shows a microscopic view at 100 x magnification. The pictures B and C made with a bigger magnification (400 x) show respectively a neutrophil population (view 1) and a neutrophil population with one eosinophil contaminant cell (view 2). Based on 5 different microscopic fields, the purity of the neutrophil population was estimated to 97.1 ± 0.9 % neutrophils. The other cells are eosinophils. This information was added in the material and method section (from line 197).         

Figure 1:  Picture of a discontinuous Percoll gradient obtained for horse neutrophils isolation according to the technique of Pycock et al. and pictures (A, B, C) of neutrophil smears obtained after neutrophil isolation and Diff Quick staining.

  1. The concentration of PMA frequently used are to 10 at 100nM, why the authors used 16uM? This concentration are elevated and could by cytotoxic.

Indeed, this concentration was diluted 20 x in presence of the cells, so the final concentration of PMA was 0.8 µM. This was stated in the paper (line 210).

  1. The authors deduce the NET’s formation by elevation of ROS production and the release of MPO but not include any image for make sure of these.

Thank you for this remark. You are right! For the next studies, it will be interesting to make an immunological staining of isolated neutrophils after their stimulation to follow MPO release and NET formation. But please, allow me to show you first pictures obtained in an inflamed tissue of horse foot during an inflammatory pathology call laminitis. The immunological staining using the same antibodies used in this study show localized or more diffuse brown staining which seems to correspond to the presence of neutrophils. There is an evid

Dear Reviewer,

I would like to thank for your comments and remarks which help to improve the paper.

We hope we have answered your questions sufficiently.

Thank you for your assistance

Thierry Franck 
